# Development and validation of real-time recombinase polymerase amplification-based assays for detecting HPV16 and HPV18 DNA

Jiaxu Ying,[1,2,3] Lingjing Mao,[1,2,3] Yujing Tang,[4] Meriem Fassatoui,[1] Wei Song,[4] Xiaosheng Xu,[4] Xiaojian Tang,[4] Jing Li,[4] Hua Liu,[4] Fangfang Jian,[4] Qinwen Du,[4] Gary Wong,[2] Weiwei Feng,[4] Nicolas Berthet[1,5,6]

**ABSTRACT**   Cervical cancer is the fourth leading cause of cancer-related death among women worldwide. Persistent human papillomavirus (HPV) infection is the principal cause of cervical cancer, with HPV16 and HPV18 accounting for about 70% of cases worldwide. Cervical screening is an effective secondary measure for preventing cervical cancer. Testing for HPV DNA is becoming increasingly important in cervical cancer screening. The existing PCR-based HPV detection technology has several disadvantages: the assays are time-consuming, and sophisticated equipment is required to control the temperature cycles. These drawbacks have led to the development of detection technologies based on isothermal amplification. Here, we present real-time recombinase polymerase amplification (RPA-exo)-based assays for single genotyping of either the E7 or the L1 segment of HPV16 or HPV18. These assays were highly sensitive, able to detect HPV in all clinical samples with Ct values below 34, and yielded results within 25 min. A dual-detection system capable of detecting both HPV16 and HPV18 in a single reaction was also developed based on the L1 gene. It has a limit of detection of approximately 10,000 copies of each genotype per reaction. The assays were validated with DNA extracted from 36 biopsy specimens and 42 exfoliated cell samples from 43 patients with cervical lesions at different stages. The RPA-exo system is a promising clinical detection platform with the advantages of yielding results rapidly and operating at a constant temperature, while being cost-effective and easy to use.

**IMPORTANCE**   HPV DNA screening is an effective approach for the prevention of cervical cancer. The novel real-time recombinase polymerase amplification-based HPV detection systems we developed constitute an improvement over the HPV detection methods currently used in clinical practice and should help to extend cervical cancer screening in the future, particularly in point-of-care test settings.

**KEYWORDS**   cervical cancer, human papillomavirus, real-time recombinase polymerase amplification

Cervical cancer is the fourth most frequently diagnosed cancer and the fourth leading cause of cancer deaths in women worldwide. According to the GLOBOCAN 2020 database, cervical cancer is the most frequently diagnosed cancer in 23 countries and the leading cause of female cancer deaths in 36 countries (1). In 2020, China reported 109,741 cases of cervical cancer and 59,060 deaths, with a corresponding age-standardized incidence of 10.7 cases per 100,000 women-years and mortality rate of 5.3 deaths per 100,000 women-years. The incidence of cervical cancer in China is currently still above the threshold set by the World Health Organization (WHO)'s Cervical Cancer Elimination Initiative, which is four cases per 100,000 women-years (2). The WHO

Address correspondence to Gary Wong, garyckwong@ips.ac.cn, Weiwei Feng, fww12066@rjh.com.cn, or Nicolas Berthet, nicolas.berthet@pasteur.fr.

The authors declare no conflict of interest.

See the funding table on p. 14.

announced the elimination of cervical cancer as a global public health priority in 2018 (3) and a worldwide strategy to achieve this end was launched in November 2020 (4). Cervical cancer development is closely related to persistent human papillomavirus (HPV) infection. Almost all cases of cervical cancer are thought to be associated with HPV, an 8 kb, circular, non-enveloped, double-stranded DNA virus (5). Fifteen types of HPV (genotypes 16, 18, 31, 33, 35, 39, 45, 51, 52, 56, 58, 59, 68, 73, and 82) are currently classified as high-risk (6), including HPV16 and HPV18, the most prevalent types worldwide. In mainland China, the overall rate of infection with high-risk HPVs (HR-HPVs) is about 19% (7).

Cervical cancer is considered to be almost entirely preventable, with highly effective primary (HPV vaccine) and secondary (screening) preventive measures (1). Cytological cervical cancer screening methods (Pap smears) can effectively identify precancerous lesions, making it possible to treat cases early. However, in developing countries, cervical cancer screening programs may be limited or non-existent due to inadequate health-care resources; as a result, most cervical cancer patients are already in the advanced stage of the disease when diagnosed (8). The sensitivity and accuracy of cytological methods are affected by many factors, including laboratory medical infrastructure, the experience of the cytologists, and the quality of sample fixation and staining (5). There is, therefore, a need for more convenient, sensitive, and cheaper screening methods yielding more consistent results. In the latest WHO guidelines for the screening and treatment of cervical pre-cancer lesions for cervical cancer prevention, HPV-DNA testing is recommended as the principal screening approach, with partial genotyping, colposcopy, or cytological methods used for triage in women with a positive HPV DNA test (9). The genotyping of HPV16 and HPV18 is recommended by the WHO as they are the genotypes most frequently implicated in patients with cervical cancer (9). A randomized multicenter clinical trial in China demonstrated that HR-HPV testing was as effective for screening as cytological methods or acetic acid and lugol iodine (VIA/VILI) staining. Furthermore, referring only women with HPV16 or HPV18 for direct colposcopy decreased the colposcopy referral rate without significantly increasing disease incidence (10).

Polymerase chain reaction (PCR) is currently the principal technique for oncogenic HPV DNA detection and identification in cervical lesion biopsy specimens and exfoliated cells. Various commercial kits based on DNA-DNA hybridization capture and real-time quantitative PCR (qPCR) have been developed and validated (11). However, these techniques require expensive specialist equipment to control the temperature cycle, and different companies always require corresponding equipment for conducting experiments with their kits and results readout, resulting in an overall increase in the cost of the tests. In clinical and field conditions, there is a demand for more convenient rapid point-of-care testing (POCT), with simpler equipment requirements, and lower costs. Novel HPV detection methods, such as droplet digital PCR (ddPCR) (12), AuNP (gold nanoparticle)-HPV probe characterization assays (13), aggregation-induced emission (AIE) combined with flow-through hybridization technologies (14), recombinase polymerase amplification (RPA)-based assays (15), loop-mediated amplification (LAMP) reactions (16), and RPA combined with CRISPR-Cas12 (17), have been developed, with the aim of shortening the time required to obtain results, simplifying the interpretation of results and reducing diagnostic costs. Until now, a total of 15 articles have applied RPA to HPV detection (Table 1).

RPA is an isothermal method for amplifying nucleic acids. It has two principle advantages: (i) RPA is performed at a constant temperature of 37–42℃, rendering sophisticated equipment, such as thermocyclers, unnecessary; and (ii) RPA reactions are fast, with the ideal level of nucleic acid amplification for detection achieved within 20–30 min (31). The TwistAmpTM exo probe is used specifically for fluorogenic real-time RPA detection (referred to hereafter as RPA-exo). RPA can tolerate mismatches, the highest mismatch tolerance reported to date being nine nucleotide base pairs between the

**TABLE 1** Recombinase polymerase amplification (RPA) detection systems developed for HPV

| Author (Year) | Principle | Clinical sample information | HPV genotype | Gene target and position (nucleotide) | Amplification temperature (°C) and time | Readout method | LOD[c] | Reference |
|---|---|---|---|---|---|---|---|---|
| Biao Ma et al. (2017) | RPA | Cervical swab samples (n = 335), using DNA extraction Kit | 16, 18 ([a]Separate genotyping) | L1, HPV16: 1063–1207, HPV18: 1168–1315 | 37°C 20 min for RPA amplification, 90 min for readout | Agarose gel electrophoresis assay, RDB[d], or quantitative real-time assay with SYBR Green I | 0.1 fg/μL (10^0 Copies/μL) | (18) |
| Janice S. Chen et al. (2018) | DETECTR, Cas12a combining RPA | Cervical swab samples (n = 25), DNA extracted by adding Tris-EDTA with proteinase K and heating | 16, 18 (Separate genotyping) | L1, HPV16: 1047–1183, HPV18: 1158–1278 | 37°C 10 min RPA amplification and 37°C 60 min Cas12a cleavage | Fluorescent signals collected every 30 s | 1 aM (approximately 0.6 copies/μL[b]) for HPV16 and 10 aM (6 copies/μL) for HPV18 | (19) |
| Biao Ma et al. (2019) | combined method of isothermal RPA, LFD[e] and RDB | Cervical swab samples (n = 450), DNA extracted by adding lysis buffer and heating | 6, 11, 16, 18, 26, 31, 33, 35, 39, 40, 42, 43, 44, 45, 51, 52, 53, 56, 58, 59, 66, 68, 73, 81, and 83 (non-genotyping for LFD, separate genotyping for RDB) | L1, HPV16: 1024–1205, HPV18: 1129–1313 | 37°C 30 min for RPA-LFD. For RDB, 51°C 30 min for RPA amplification | Visual observation of LFD | 100 fg (100 copies) for LFD and 1,000 fg (1,000 copies) for RDB[d] | (15) |
| Jen-Hui Tsou et al. (2019) | Cas12a combining RPA with LFD readout | Plasma samples (n = 29), DNA extracted by adding phosphate-buffered saline with 0.2% Triton X-100 and heating | 16, 18 (Separate genotyping) | L1, HPV16: 1047–1183, HPV18: 1158–1278 | 37°C 20 min RPA amplification, 37°C 180 min Cas12a cleavage and room temperature 5 min | Visual observation of LFD | 0.28 fM (170.6 copies/μL) for HPV16 and 0.24 fM (151.8 copies/μL) for HPV18 | (20) |
| Kun Yin et al. (2020) | RPA combined with CRISPR-Cas12a detection, supported by dynamic aqueous multiphase reaction system | Cervical swab samples (n = 15), using DNA extraction Kit | 16, 18 (Separate genotyping) | L1, HPV16: 1047–1183, HPV18: 1158–1278 | 37°C 60 min | Analyze the intensity of endpoint fluorescence | 10 copies for HPV16 and 100 copies for HPV18 | (21) |
| Yi Wang et al. (2020) | RPA assisted with graphene oxide and self-avoiding molecular recognition systems, LFD also developed | Samples collected after clinical diagnosis (n = 89), using DNA extraction Kit the type of samples collected was not indicated | 16, 18 (Genotyping in 1 reaction) | L1, HPV16: 913–1042, HPV18: 1044–1261 | 38°C 30 min | Visual observation of LFD | 10 copies for HPV16 and HPV18 | (22) |
| Jiaojiao Gong et al. (2021) | RPA combined with CRISPR-Cas12a | Cervical swab samples (n = 6), using DNA extraction Kit | 16, 18, 31, 33, 35, 39, 45, 51, 52, 56, 58, 59, 68 (non-genotyping) | L1, MY11/GP6 + primer target | 37°C 20 min RPA amplification and 37°C 60 min Cas12a cleavage | Fluorescent signals collected every 30 s | 500 copies for all HR-HPVs | (17) |

*(Continued on next page)*

**TABLE 1** Recombinase polymerase amplification (RPA) detection systems developed for HPV (Continued)

| Author (Year) | Principle | Clinical sample information | HPV genotype | Gene target and position (nucleotide) | Amplification temperature (°C) and time | Readout method | LOD$^c$ | Reference |
|---|---|---|---|---|---|---|---|---|
| Phetploy Rungka moltip et al. (2021) | RPA combined with LFD | Serum specimens (n = 73), cell-free DNA extracted kit used, 12 samples used for RPA-LFD assay | 16, 18 (Separate genotyping) | E7 HPV16: 106–191 HPV18: 204–273 | 37°C 30 min RPA and 7 min for visualization | Visual observation of LFD | 10 copies/µL for HPV16 and 5 copies/µL for HPV18 | (23) |
| Ziyue Li et al. (2021) | Electrochemical CRISPR-Cas12a biosensor (Combining RPA when testing clinical samples) | Cervical swab samples (n = 6), using DNA 16 extraction Kit | 16 | L1 1047–1183 | 37°C 10 min RPA amplification and 37°C 60 min Cas12a cleavage | Analyze the intensity of endpoint fluorescence | 1 pM without RPA ($10^5$ copies/µL), LOD with RPA was not indicated | (24) |
| Tao Hu et al. (2022) | Chemiluminescence enhanced CRISPR-Cas12a biosensor combined with RPA | Cervical swab samples (n = 20), DNA extraction method was not indicated | 16 | L1 270–476 | 37°C 30 min RPA amplification, 37°C 30 min Cas12a cleavage, 10 min 0.2% SDS to destroy the activity of Cas12a, 60 min room temperature for signal amplification, 10 min for signal readout | Chemiluminescence signal measured or visually read and interpreted | 1 copy/µL | (25) |
| Zhichen Xu et al.(2022) | CRISPR-Cas12a and multiplex RPA on a microfluidic device | Cervical cell specimens (n = 100); heated at 95°C for 10 min | 6, 11, 16, 18, 31, 33, 45, 52, 58 (Separate genotyping) | L1 Different for each genotype, within the 5571–6871 region | 39°C 20 min RPA amplification and 37°C 15 min Cas12a cleavage | Analyze the intensity of endpoint fluorescence | 0.26 aM (approximately 0.16 copies/µL$^b$) | (26) |
| Jinjoo Han et al.(2023) | Blocker DNA-assisted Cas12a combined RPA | Cervical tissue samples (n = 32); using gDNA extraction Kit | 16, 18 (Genotyping in 1 reaction) | L1 HPV16: 92–168 HPV18: 1624–1703 | 37°C 10 min RPA, 95°C 5 min for inactivation, 37°C 30 min transcription, 80°C 10 min for inactivation, 37°C 15 min Cas12a cleavage | Fluorescence signals collected every minute | 1 aM (approximately 0.6 copies/µL) | (27) |
| Hu Zhou et al. (2023) | RPA combined Cas12 assisted by a heating-membrane- multiplexed microfluidics platform | Cervical cell specimens (n = 32); heated at 95°C for 10 min | 16, 18 (Separate genotyping) | L1 HPV16: 1024–1207 HPV18 L1: 1129–1315 | 39°C 12 min RPA amplification and 37°C 12 min Cas12a cleavage | Visual observation of LFD$^e$ | $1 \times 10^{-18}$ M (approximately 1 copies/µL$^b$) | (28) |
| Yin Zhao et al. (2023) | CRISPR-Cas12a combined with multiplexed | Cervical cell specimens (n = 20); heated at 95°C for 10 min | 16, 18 (Genotyping in 1 reaction) | L1 HPV16: 122–408 | 39°C 20 min RPA amplification | Visual observation of bright-field images and fluorescent images | $1 \times 10^{-18}$ M (1 copy/reaction) | (29) |

**TABLE 1** Recombinase polymerase amplification (RPA) detection systems developed for HPV (*Continued*)

| Author (Year) | Principle | Clinical sample information | HPV genotype | Gene target and position (nucleotide) | Amplification temperature (°C) and time | Readout method | LOD[c] | Reference |
|---|---|---|---|---|---|---|---|---|
| | RPA on a simple microfluidic dual-droplet device | | | HPV18: 685–1088 | and 37°C 25 min Cas12a cleavage | | | |
| Rungda wan Wongsa mart et al. (2023) | Multiplex RPA assay | Cervical swab samples (n = 130), DNA were extracted automatically using the Cobas 4800 system | 20 HPV HR types (16, 18, 26, 31, 33, 35, 39, 45, 51, 52, 53, 56, 58, 59, 66, 68, 69, 73, 82) and 14 LR types (6, 11, 32, 40, 42, 43, 44, 54, 61, 70, 72, 81, 84, 87) (non-genotyping) | L1, E6 Different for each genotype, around 1150–1345 for L1, and 130–260 for E6 | 39°C 40 min, 80°C heating for 2 min | DNA bands separated by 3% agarose gel electrophoresis | 2.01 fg (1,000 copies) for L1 and 0.0125 fg (100 copies) for E6/E7 Only HPV11,18,33 tested for LOD | (30) |
| This article | Real-time RPA Assay | Cervical swab samples (n = 42), cervical biopsies (n = 36), DNA extracted by phenol-chloroform | 16, 18 (Genotyping in 1 reaction) | L1, E7 HPV16 L1: 1024–1205 HPV18 L1: 1129–1313 HPV16 E7: 87–269 HPV18 E7: 143–308 | 37°C 25 min | Fluorescence signals collected every 15 s | 1,000 copies/µL for HPV16 L1, 500 copies/µL for HPV16 E7, 50 copies/µL for HPV18 L1 and 100 copies/µL for HPV18 E7; 10,000 copies for genotyping in 1 reaction | |

[a]Separate genotyping indicates that the presence of different genotypes cannot be obtained in one reaction tube.
[b]The authors did not provide a detection limit converted to virus copy number, the viral copies were estimated according to the length of plasmid used.
[c]Limit of detection.
[d]RDB, reverse dot blot.
[e]LFD, lateral flow dipstick.

primer and probe binding sites, making it possible to design degenerate or consensus primers and probes for detecting different pathogen subtypes in multiplex assays (32).

HPV classification is based on the highly conserved L1 gene. The L1 open reading frame of each HPV type differs from that of any other HPV type by at least 10%, the target for most genotyping methods (33). However, the E6 and E7 proteins are more closely related to cell transformation, and their dysregulation often leads to cellular genomic mutations; the detection of these two genes may, therefore, be more clinically meaningful for evaluating disease progression (8). Several studies have shown that the L1 interval is more likely to be interrupted during HPV integration than the interval containing E6 or E7 (34), potentially resulting in false-negative results. In this study, we developed RPA-exo-based assays for single genotyping based on either L1 or E7 for HPV16 or HPV18. The limit of detection (LOD) within 25 min for each assay is: 1,000 copies/µL for HPV16 L1, 500 copies/µL for HPV16 E7, 50 copies/µL for HPV18 L1 and 100 copies/µL for HPV18 L1. We also developed an assay for the simultaneous RPA-exo-based detection of HPV16/18 L1 in a single reaction tube. We designed degenerate primers, which we used with HPV16/18-specific probes for L1, for the simultaneous amplification of both genotypes for detection purposes. The dual-detection system had a LOD of 10,000 copies/µL for either HPV16 or HPV18. We validated these five assays with cervical samples from 43 patients and compared the results with those obtained with conventional qPCR methods.

## RESULTS

### Development of different detection methods

Tenfold serial dilutions of HPV16 and HPV18 L1 plasmids were used for the generation of qPCR standard curves (Fig. 1A and B). For L1 plasmids, qPCR had a LOD of 10 copies/µL for both HPV16 and HPV18, with a coefficient of correlation of 0.999. These qPCR tests were then used as the gold standard for comparisons with the results of RPA-exo assays. The analytical sensitivity of single RPA-exo assays was evaluated also by testing 10-fold serial dilutions of the various plasmids. We developed five RPA-exo assays in total: single HPV16 E7 RPA-exo, single HPV16 L1 RPA-exo, single HPV18 E7 RPA-exo, single HPV18 L1 RPA-exo, and a dual-detection RPA-exo assay for HPV16 and HPV18 L1. All single RPA-exo assays were specific and the LOD within 25 min for each assay is: 1,000 copies/µL for HPV16 L1, 500 copies/µL for HPV16 E7, 50 copies/µL for HPV18 L1, and 100 copies/µL for HPV18 L1 (Fig. 1C). The specificity of the E7 RPA-exo and HPV16 L1 detection systems was assessed with $10^4$ copies/µL of plasmids of different genotypes. For example, for HPV16 E7 RPA-exo detection, we used $10^4$ copies/µL of the HPV18 E7 plasmid to check specificity, and DNA from SiHa cells, which contains integrated HPV16 DNA, was used to validate specificity for HPV18 L1 RPA-exo (Fig. 1D). We optimized several key parameters of the assay to ensure the best possible detection performance for the simultaneous identification of L1 from both HPV16 and HPV18. The concentration of the primers was increased from 420 to 480 nM, and probe concentrations of 120 nM for HPV16 and 100 nM for HPV18 were used. The results of the dual RPA-exo assay showed both RPA-exo probes have good specificity for the L1 segment, as HPV16 DNA was not detected with the HPV18 probe, and vice versa. Relative to single-genotype detection, some loss of sensitivity was observed when HPV DNA was amplified with degenerate primers. However, at least for amounts of input plasmid DNA of up to 10,000 copies/µL, genotype detection was positive regardless of the presence of another HPV genotype (Fig. 1E).

### Validation of assay performance on clinical samples

The 36 biopsy specimens and 42 ThinPrep cytological test (TCT) samples are tested to validate the utility of RPA-exo assays. Four samples (79, 81, 144, and 175) marked to be co-infected with HPV16 and HPV18 by hospital, so these samples were counted two times. Using the infection information provided by the hospital, we first reconfirmed the genotype of HPV present in all extracted DNA samples by traditional qPCR, and we then

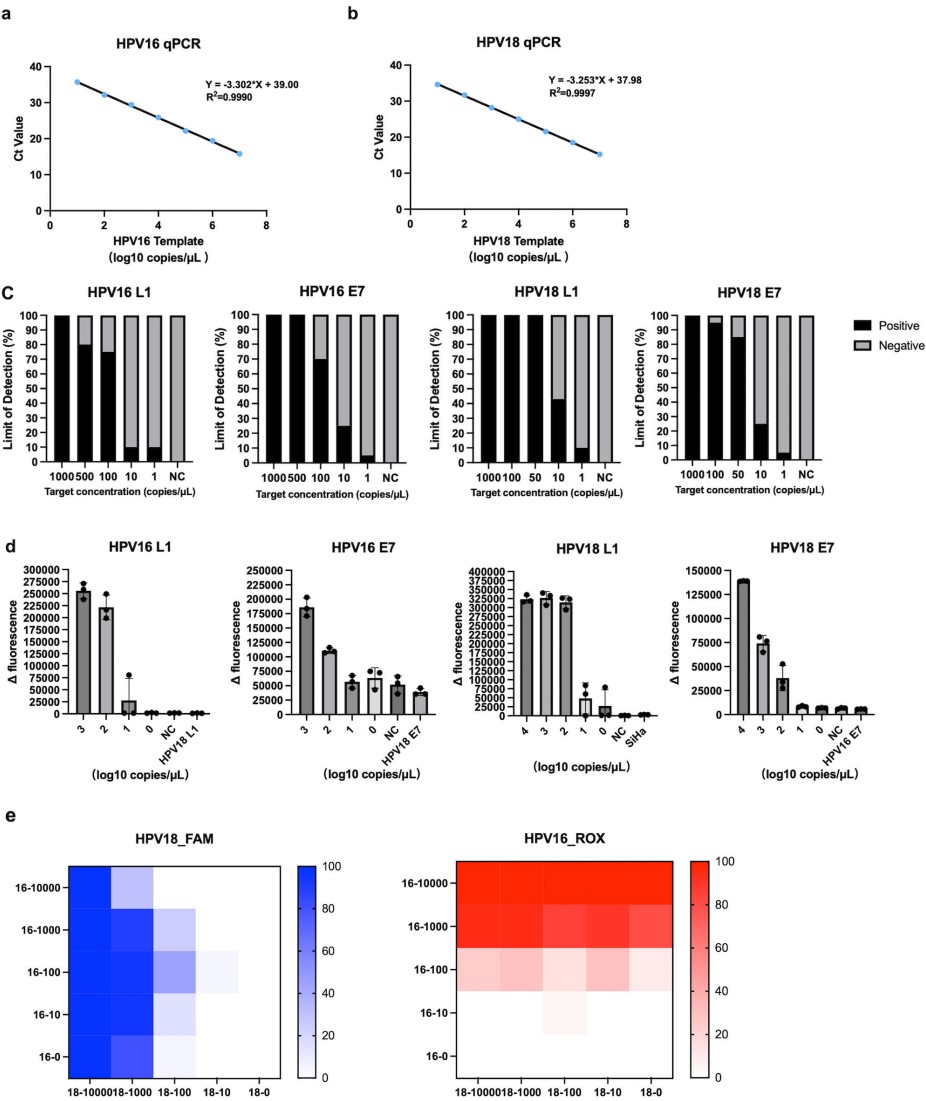

**FIG 1** Development of different assays and their performance. (A and B) Tenfold serial dilutions ranging from $10^7$ to $10^1$ copies of HPV16 and HPV18 plasmid. A standard curve is generated by plotting the CT values on the $y$ axis and the log of viral DNA copies per reaction on the $x$ axis. (C) Sensitivity and LOD evaluation of RPA-exo assays for L1 and E7. Determine the LOD by elucidating the lowest concentration in which the HPV DNA template could be detected at or above 95% over 20 replicates in 20 min. (D) Specificity of RPA-exo assays for L1 and E7. Except for HPV18 L1 RPA-exo we used DNA extracted from SiHa, the specificity of other systems was assessed with $10^4$ copies/µL plasmids of the same gene region in different genotypes. (E) Sensitivity of dual RPA-exo assays. The heatmap shows the positive rate of HPV16 and HPV18 plasmids mixture under different concentration.

tested the samples for each HPV genotype separately. All assays were tested in triplicate, with all samples (Table S2 and S3). Specificity relative to qPCR was 100% for all the assays tested, and the sensitivity of each assay is reported in Table 2. In general, sensitivity was better for detection of the L1 gene than for E7. HPV16 L1 single RPA-exo was able to detect HPV with a Ct value below 34.3, except for sample 10, for which negative results were obtained (we repeated HPV16 L1 single RPA-exo for this sample and obtained similar results). The HPV18 L1 single RPA-exo detects HPV with a Ct value below 34.4. Positive RPA-exo results were obtained for all samples with a Ct value corresponding to over 100 copies/µL according to the standard curve for qPCR, confirming the LOD. The overall sensitivity of the dual-detection system was 86.2% (56/65). In comparisons of the

**TABLE 2** Sensitivity and specificity of RPA-exo assays for the detection of HPV16 and HPV18 in biopsy and TCT samples[a]

| HPV16 L1 RPA-exo Result | Samples testing positive by qPCR (n = 54) | Samples testing negative by qPCR (n = 8) | Sensitivity | Specificity |
|---|---|---|---|---|
| Positive | 48 | 0 | 48/54 (88.9%) | |
| Negative | 6 | 8 | | 8/8 (100%) |
| **HPV16 E7 RPA-exo Result** | **Samples testing positive by qPCR (n = 54)** | **Samples testing negative by qPCR (n = 8)** | **Sensitivity** | **Specificity** |
| Positive | 46 | 0 | 46/54(85.2%) | |
| Negative | 8 | 8 | | 8/8 (100%) |
| **HPV18 L1 RPA-exo Result** | **Samples testing positive by qPCR (n = 11)** | **Samples testing negative by qPCR (n = 11)** | **Sensitivity** | **Specificity** |
| Positive | 9 | 0 | 9/11(81.8%) | |
| Negative | 2 | 11 | | 11/11 (100%) |
| **HPV18 E7 RPA-exo Result** | **Samples testing positive by qPCR (n = 11)** | **Samples testing negative by qPCR (n = 11)** | **Sensitivity** | **Specificity** |
| Positive | 8 | 0 | 8/11 (72.7%) | |
| Negative | 3 | 11 | | 11/11 (100%) |
| **Dual RPA-exo Result** | **Samples testing positive by qPCR (n = 65)** | **Samples testing negative by qPCR (n = 19)** | **Sensitivity** | **Specificity** |
| Positive | 57 | 0[b] | 57/65 (87.7%) | |
| Negative | 8 | 19 | | 19/19 (100%) |

[a]The co-infected samples were counted twice.
[b]TCT sample (T94) was found to be co-infected with HPV16 and HPV18 in the dual-detection assay, despite the hospital notes indicating that the sample was positive only for HPV18. The Ct value for HPV16 qPCR was 36.92, and a positive result was also obtained with the HPV16 RPA exo assay for L1.

detection of the two genotypes separately, sensitivity was 88.9% (48/54) for HPV16 detection and 72.7% (8/11) for HPV18 detection. For all HPV16 samples, the dual assay detected HPV with Ct values less than 33.7. For all HPV18 samples, HPV was detected with a Ct below 32.1. In general, the dual-detection assay conserved a high sensitivity for HPV16-positive clinical samples, but was less sensitive than single RPA-exo detection for HPV18, at 72.7% (8/11) as opposed to 81.8% (9/11). There was also good agreement in clinical samples using other DNA extraction methods, validating the applicability of these assays for DNA with multiple extraction methods (Table S4).

## DISCUSSION

With advances in HPV vaccination, programs for eliminating cervical cancer are well underway and have successfully reduced the incidence of cervical cancer in high-income countries. However, there have been worldwide shortages of HPV vaccines and few regions of the world have vaccination rates close to the ideal, especially in Asia and Africa (35). The nine-valent HPV vaccine was not approved in China until 2018, and vaccine coverage remains low or non-existent, particularly in resource-poor areas, due to the low availability and high price of the vaccine (36). None of the patients studied here were vaccinated. High-quality screening technology is still required for the sorting and management of at-risk groups, to ensure the early detection and treatment of cervical cancer, to cure reversible precancerous lesions, thereby decreasing mortality.

HPV testing as the primary screening tool for cervical cancer has several clear advantages over standard cytological tests, including higher throughput, greater sensitivity, and better reproducibility. The PCR-based detection method is currently the gold standard for HPV DNA detection in clinical practice, but faster, more convenient methods need to be developed, to overcome disparities in medical resources. High-quality RPA tests have been developed for the detection of many viruses, including HPV (20, 37), the Zika, and Dengue viruses (38), HIV (39), yellow fever virus (40), and monkeypox virus (41). For RPA-based HPV detection assays, more than half of the systems are still combined with Cas12 (out of 15 assays, 9 used Cas12, Table 1), and technical

aspects of one-step multiplex testing remain to be optimized. In addition, most of these assays (8 out of 15 assays, Table 1) can only achieve separate genotyping, which means that detection of different genotypes still needs to be achieved in different reaction systems, and multiple sets of primers or probes cannot be realized in the one reaction. While genotyping in the same reaction has the advantages of saving reagent costs and reducing duplicate addition of samples. Efforts are therefore being made to develop POCT methods that can identify these two genotypes simultaneously, based, for example, on the combination of PCR and a lateral flow strip (LFS) for the visual typing of HPV16/18 (42).

In this study, we developed assays for the E7 and L1 gene segments of the two most common HPV genotypes with sensitivity of and below 1,000 copies/µL, and a dual-detection assay with a sensitivity of 10,000 copies/µL for each genotype. The observed LOD in tests with clinical samples was actually slightly lower than 100 and 1,000 copies/µL, as demonstrated by comparison with qPCR results. As stated above, the HPV16 L1 single RPA-exo assay detected viral DNA with a Ct value below 34.3, equivalent to 26.5 copies/µL, according to the standard curve. The HPV18 L1 single RPA-exo assay detected viral DNA with a Ct value below 34.4, equivalent to 12.6 copies/µL. The dual-detection system was also able to detect Ct value as low as 34.3 for HPV16 (patient 106's TCT) and 34.4 for HPV18 (patient 68's TCT). The presence of more than 10 copies/µL of HPV16 or HPV18 genome is considered clinically significant, and lower values probably reflect transient infections (43). Considering the detection section alone, our assay has advantages of easy primer/probe design, quick operation, short incubation time, constant and low incubation temperature, and genotyping in one reaction. Due to the lack of DNA templates for other HR-HPVs, we only used HPV18 DNA when verifying the specificity of HPV16 detection systems, and vice versa. The single genotype assays were all considered to be highly specific, as specific primers and probes were used. For the dual-detection system, the alignments showed that our probes targeted the highly divergent region (Fig. 2). As shown, the degenerate primers had the potential to amplify other genotypes, but the specific probes eliminated the possibility of off-target detection, as the region to which the probes bind was not conserved. The designed probe had <90% nucleotide sequence similarity with any other HR-HPV types in the interval, thus ensuring the accurate identification of the corresponding HPV type. When we used this system to detect samples that were only HPV16 positive, there was no fluorescence detected from the HPV18 probe (and vice versa), demonstrating the specificity of the dual-detection assay. We are, therefore, confident that this method is of potential value for use in clinical applications.

When the E7 protein sequence was used to construct a phylogenetic tree, it was found that oncogenic or possibly oncogenic HPVs clustered together. As the E7 expressed by HR-HPVs can degrade pRb, an important tumor suppressor factor, we thought that the use of E7 as the target gene might be more clinically meaningful (44). As the target gene for E7 RPA-exo assays was different from the qPCR target, the results were less consistent. For example, all the samples from patient 153 (T153 and B153) and the TCT sample from patient 167 (T167) gave positive results for L1 detection, but the E7 tests were negative, whereas the opposite pattern was observed for the biopsy specimen from patient 79 (B79). This finding indicates that for the same sample, the detection results for different gene regions of HPV may differ, and the clinical detection of only one gene segment potentially resulting in false-negatives results. This phenomenon may be related to the diverse physical states of HPV during its life cycle, as the viral DNA may be episomal, integrated or both. However, we found that background levels were higher of our E7 detection system than for the L1 detection system, and the high fluorescence values in the NC made it difficult to define the threshold. E7 detection may not, therefore, be the best choice when using the RPA-exo system.

Whilst testing the ability of our qPCR and RPA-exo systems to detect HPV in clinical samples, we found that some samples labeled as HPV-infected gave negative results with our methods. This problem affected HPV18-infected samples more than HPV16-infected

samples, and biopsy specimens more than TCT samples. For example, the proportion of samples testing negative for HPV16 was 17.9% (5/28) for biopsy specimens, but only 8.9% for TCT samples (3/34). Unfortunately, the Ct value of all the samples was not available from our collaborators at the hospital. The discrepancies between our results and those obtained at the hospital may be due to differences in the precise TCT and biopsy samples tested between our laboratory and the hospital. There may have been errors in the choice of sampling site based on examination with the naked eye, and the biopsy tissues obtained may not therefore have included the lesion and may not have been infected with HPV, resulting in a higher proportion of negative results for biopsy specimens. The repeated freezing and thawing of the sample may also have led to a degradation of the DNA, giving rise to false-negative results.

According to the information provided by the hospital, 15 of the 43 patients were infected with multiple genotypes of HPV. In addition to four patients with HPV16/18 co-infection, there were nine patients infected with HPV16 plus another genotype of HPV (genotypes 31, 33, 51, 52, 53, and 58) and two patients infected with HPV18 plus another genotype of HPV (genotypes 58 and 66). Sample 94 emitted both ROX and FAM fluorescence during dual detection (and was, thus, positive for both HPV16 and HPV18), but the information supplied by the hospital mentioned only positivity for HPV18. We validated the co-infection of this sample with HPV16 by qPCR and single-genotype RPA-exo also targeting the L1 gene, confirming the existence of a flaw in current clinical screening practices.

The kit used to genotype 15 HR-HPVs from TCT samples at the hospital costs ¥300 (US$43.9), whereas our assays cost only about ¥55 ($8) for genotyping of the two most frequent HPV genotypes. Testing time was also substantially reduced, from 90 to less than 30 min.

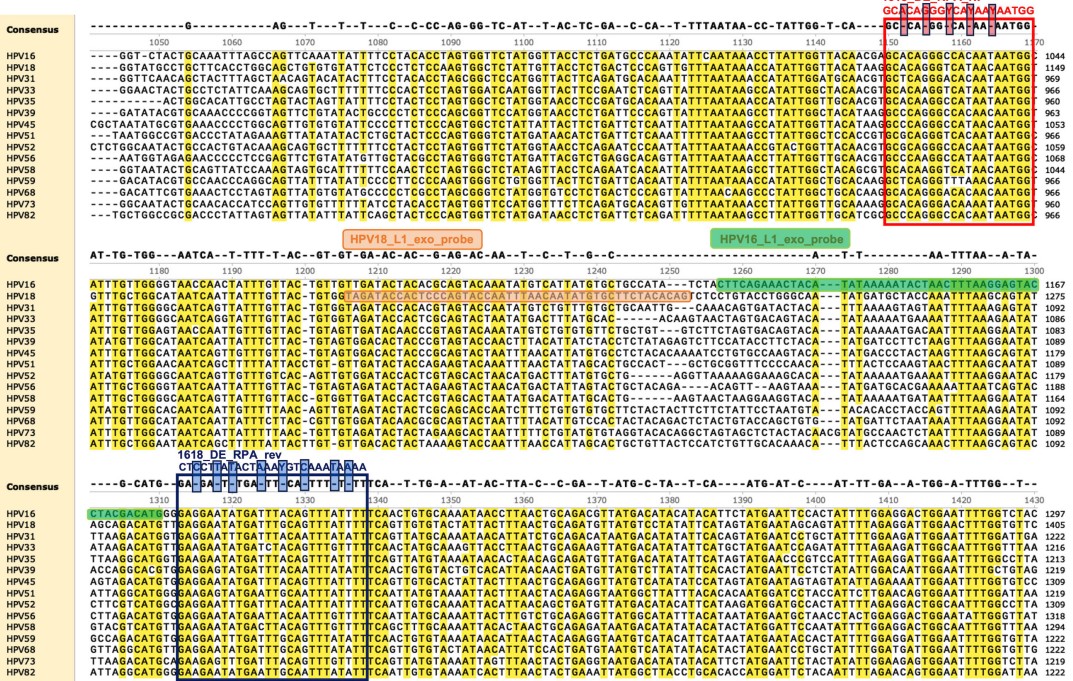

**FIG 2** Alignment of partial L1 sequences for 15 HR-HPVs. The GenBank accession numbers for the sequences used are as follows: K02718.1 (HPV16), X05015.1 (HPV18), J04353.1 (HPV31), M12732.1 (HPV33), X74477.1 (HPV35), LR861938.1 (HPV39), X74479.1 (HPV45), M62877.1 (HPV51), X74481.1 (HPV52), X74483.1 (HPV56), D90400.1 (HPV58), X77858.1 (HPV59), DQ080079.1 (HPV68), X94165.1 (HPV73), and AB027021 (HPV82). The alignment was performed by the Geneious Prime program and shown in SnapGene. The consensus sequence with threshold of >90% is shown in yellow highlight. The degenerate primers and specific probes (modifications not marked) for dual RPA-exo detection are indicated by different colored boxes. The forward primer is shown above the red box, and the reverse primer is shown above the blue box. When choosing degenerate bases, only HPV16 and HPV18 genotypes were considered, marked with small boxes of corresponding colors. The HPV16 L1 RPA exo probe is shown in the green box while the HPV18 L1 RPA exo probe is shown in the orange box.

This study had several limitations, including its small sample size, the use of only female cervical samples (no plasma samples), did not use other HPV genotypes for specificity verification and tedious phenol-chloroform sample extraction method. However, this detection system is applicable to all DNA samples, which means commercial kits can be used for DNA extraction as well. The conditions required for these assays are simpler than those for current tests, as a constant temperature of 37°C is used (there is therefore no need for a temperature cycling machine), but an instrument for the collection of fluorescence is still required. Alternative readout methods, such as LFS, which have been investigated for other pathogens (20, 39, 41), could be used to solve this problem. HPV-based self-sampling kits would significantly increase participation in screening programs, making it possible to reach more patients than standard clinician-based modes of sampling, which may be infeasible or unacceptable to some women (45). We are working toward the development of a complete home-sampling and testing kit, for genuine POCT, to increase the coverage for HPV testing in China. Our method provides only a qualitative analysis and cannot be used to determine HPV viral load quantitatively. However, the determination of viral load is significant in predicting invasive tumor risk (46). Other aspects of the development of new detection technologies should also be considered. The development of liquid biopsy techniques may provide opportunities for minimally invasive or even non-invasive detection methods for the early detection of HPV-associated tumors (47). Improvements in the efficacy of extraction for circulating viral tumor DNA, together with RPA-exo assays, should render cervical cancer screening feasible in areas with scarce medical resources, due to the use of more accessible types of samples and faster screening methods. HPV DNA tests can provide information only about the presence or absence of the viral DNA. They cannot indicate the cytological category of the lesion—high-grade squamous intraepithelial lesion (HSIL) or low-grade squamous intraepithelial lesion (LSIL)—or its severity. The next phase in the development of HPV testing will require a more detailed understanding of viral oncogenesis and the clinical consequences of HPV infection, such as HPV integration-related gene expression (48) or methylation variations that can be used as molecular panels for predicting cervical cancer risk and disease progression. Methods comparing E2 and E6 mRNA ratios as a means of assessing the degree of integration can be used as a reliable predictor of CIN2+ (48). The achievement of those goals will require next-generation sequencing (NGS), which is increasingly used in HPV studies, not only to detect multiple targets, but also to provide information about virus-host interactions (49). Following the release in 2019 of Cecolin, the first Chinese domestic bivalent vaccine against HPV16 and HPV18, vaccination rates are expected to increase and the combination of HPV DNA screening once every 5 years with vaccination should be a cost-effective strategy for preventing cervical cancer in China (50).

In conclusion, we have successfully developed several RPA-exo assays for detecting the E7 and L1 regions of HPV16 and HPV18 with high specificity and sensitivity, and a dual HPV16/18 assay based on the L1 region. Both the single and dual-detection assays are suitable for application in clinical practice, potentially decreasing the cost of cervical cancer screening and reducing the burden placed on cytologists.

## MATERIALS AND METHODS

### Sample collection

We collected 36 biopsy specimens and 42 TCT samples from 43 patients at Ruijin Hospital, from August 2021 to August 2022 (Table S1). It was not possible to obtain biopsy tissues from seven patients, and no TCT sample was available for one patient. Patients were first screened by visual inspection with VIA/VILI staining. Patients with a positive or suspicious result underwent cytological screening. Exfoliated cells were rinsed with 20 mL of ThinPrep PreserveCyt (Hologic, Marlborough, MA, USA) fixing solution immediately after removal of the brush from the cervix and the samples were stored at 4°C (51). These patients underwent colposcopy examination and two biopsy specimens

were collected from adjacent sites in each patient. One of those specimens was sent to the pathologist for the determination of cervical intraepithelial neoplasia stage, and the other was stored overnight in 4°C in a Nunc tube (Thermo Fisher Scientific, Waltham, MA, USA) containing RNAlater (Invitrogen, Carlsbad, CA, USA). Once the tissue was fully infiltrated with RNAlater, the samples were transferred to the Institut Pasteur of Shanghai, where they were frozen and stored at −20°C. Patients were assigned categories on the basis of pathological results for the biopsy tissue, for one TCT sample subjected to HPV genotyping with Liferiver hrHPV genotyping kit (ZJ Bio-Tech, Shanghai, China) in the clinical laboratory. This study was approved by the ethics committee of Ruijin Hospital in 2021 (approval no. 74). All participants signed the informed consent form, and the study was performed in accordance with the Declaration of Helsinki (52).

## Sample preparation and DNA isolation

All samples were subjected to the following pretreatment before DNA extraction. Exfoliated cells were first subjected to centrifugation at 13,000 × $g$ for 5 min at 4°C, and the cell pellet was resuspended in 2 mL phosphate-buffered saline (PBS) (Gibco, Carlsbad, CA, USA). The cells were then washed three times in PBS, with centrifugation at 800 × $g$ for 5 min at 4°C. After the final wash, the cells were resuspended in 1 mL PBS, frozen, and stored at −80°C. Biopsy specimens were cut into pieces and the RNAlater in which they had been stored was removed by two washes in PBS with centrifugation at 800 × $g$ for 5 min just before extraction.

DNA was isolated by the traditional phenol-chloroform method and quantified with the Qubit dsDNA High-Sensitivity Assay and a Qubit 2.0 fluorometer (Life Technologies, Carlsbad, CA, USA). It was then stored at −20°C until use. DNA of 20 TCT samples was also extracted by Thermo GeneJet Viral DNA and RNA Purification Kit (Thermo Fisher Scientific) following the manufacturer's instructions. Crude DNA preparation was used for three TCT samples. The cells were suspended in 100 µL lysis solution (Thermo Fisher Scientific) with proteinase K and incubated at 56°C for 15 min, then the proteinase K was heat inactivated. Four different recombinant pUC57 plasmids containing the target regions of HPV16 L1, HPV16 E7, HPV18 L1, and HPV18 E7 were purchased from Sangon (Shanghai, China) and subjected to serial 10-fold dilution for positive controls.

## Detection by qPCR

The NovoStart Probe qPCR SuperMix (Novoprotein, Suzhou, China) kit was used for all qPCR tests. The primers and probes used for qPCR and RPA-exo assays are indicated in Table 3. Reactions were conducted in a volume of 20 µL, in accordance with the manufacturer's instructions. The cycling conditions used for the qPCR assays were as follows: denaturation at 95°C for 5 min, followed by 40 cycles of amplification at 95°C for 15 s and at 60°C for 1 min. Fluorescence signals were collected with QuantStudio1 (ABI, New York, NY, USA). Amplification of a fragment of the β-globin gene (53) was used to assess the quality of the target DNA. Specimens with negative results for β-globin gene amplification were excluded from the analysis. Each sample was run in triplicate.

## RPA-exo detection

RPA-exo probes for fluorogenic detection were designed according to the RPA guidelines of TwistDx (Cambridge, UK) and were synthesized by Sangon. The highly divergent GP5+/GP6+ region of HPV was selected as a target for the design of degenerate primers and specific probes (Fig. 2). RPA-exo was performed with the TwistAmp Exo Kit (TwistDx). For the detection of a single HPV genotype, 29.5 µL rehydration buffer, 2.1 µL forward primer (10 µM), 2.1 µL reverse primer (10 µM), 0.6 µL probe (10 µM), and 12.2 µL nuclease-free water (Invitrogen) were combined to form a master mix. For the dual-detection of HPV16 and HPV18, the master mix consisted of 29.5 µL rehydration buffer, 2.4 µL forward primer (10 µM), 2.4 µL reverse primer (10 µM), 0.5 µL HPV18 RPA-exo L1 probe (10 µM), 0.6 µL HPV16 RPA-exo L1 probe (10 µM), and 11.1 µL nuclease-free

**TABLE 3** Sequences of primers and probes for the different assays used in this study[a]

| Name | Sequences (5′ to 3′) | Assays |
|---|---|---|
| HPV18_qP_probe | 6-FAM-CAACACCTAAAGGCTGACCACGG-BHQ1 | HPV18 qPCR (54) |
| HPV18_qP_fw | TGTGCTGGAGTGGAAATTGG | |
| HPV18_qP_rev | GGCATGGGAACTTTCAGTGTC | |
| HPV16_qP_probe | 6-FAM-TGACCACGACCTACCTCAACACCTACACAGG-BHQ1 | HPV16 qPCR (53) |
| HPV16_qP_fw | CAGATACACAGCGGCTGGTTT | |
| HPV16_qP_rev | TGCATTTGCTGCATAAGCACTA | |
| HPV18_L1_exo_probe | TAGATACCACTCGCAGTACCAATTTAACAA/i6FAMdT//idSp//iBHQ1dT/GTGCTTCTACACAGT-BLOCK | HPV18 single L1 and dual RPA-exo |
| HPV18_L1_RPA_fw | TTACATAAGGCACAGGGTCATAACAATGGTGTTTG | HPV18 L1 RPA-exo |
| HPV18_L1_RPA_rev | AAACTGCAAATCATATTCCTCAACATGTC | |
| HPV16_L1_exo_probe | CATGTCGTAGGTACTCCTTAAAGTTAGTATT/iROXdT//idSp//iBHQ2dT/ATATGTAGTTTCTGAAG-BLOCK | HPV16 single L1 and dual RPA-exo |
| HPV16_L1_RPA_fw | TTGTTGGGGTAACCAACTATTTGTTACTGTT | HPV16 L1 RPA-exo |
| HPV16_L1_RPA_rev | CCTCCCCATGTCGTAGGTACTCCTTAAAG | |
| 1618_DE_RPA_fw | GCACAGGGYCAYAAYAATGG | Dual RPA-exo |
| 1618_DE_RPA_rev | AAAATAAACTGYAAATCATATTCCTC | |
| HPV18_E7_exo_probe | GTGTATGTGTTGTAAGTGTGAAGCCAGAAT/iROXdT/GA/idSp/C/iBHQ2dT/AGTAGTAGAAAGCTC- BLOCK | HPV18 E7 RPA-exo |
| HPV18_E7_RPA_fw | ATTTACCAGCCCGACGAGCCGAACCACAAC | |
| HPV18_E7_RPA_rev | GATGCACACCACGGACACACAAAGGACAG | |
| HPV16_E7_exo_probe | TCCAAAGTACGAATGTCTACGTGTGTGCTT/i6FAMdT//idSp//iBHQ1dT/ACGCACAACCGAAGC-BLOCK | HPV16 E7 RPA-exo |
| HPV16_E7_RPA_fw | TGACAGCTCAGAGGAGGAGGATGAAATAGA | |
| HPV16_E7_RPA_rev | ACAATTCCTAGTGTGCCCATTAACAGGTCT | |
| beta-471_probe | 6-FAM-TCTACCCTTGGACCCAGAGGTTCTTTGAGT-BHQ1 | β-globin qPCR |
| beta-403_fw | TGGGTTTCTGATAGGCACTGACT | |
| beta-532_rev | AACAGCATCAGGAGTGGACAGAT | |

[a]/i6FAMdT/, Int 6-FAM-dT; /iROXdT/, Int ROX-dT; /idSp/, tetrahydrofuran residue; /iBHQ1dT/, Int BHQ1 dT; /iBHQ2dT/, Int BHQ2 dT. BLOCK is the polymerase extension blocking group, in which the C3 Spacer was used.

water. The master mix was added to a 0.2 mL tube containing a dried enzyme pellet. Following the addition of 1 µL DNA template to the master mix tube, 2.5 µL magnesium acetate (280 mM) was deposited on the inside of the tube lids, and the tube was then closed, centrifuged briefly and its contents mixed well by vortex. All manipulations were performed on ice. The reaction mixture was rapidly placed in a MiniAmp thermocycler at 37°C and incubated for 4 min. Remove tubes after 4 min, vortex and spin down briefly. It was then incubated in a QuantStudio1, with fluorescence signals collected every 15 s for 80 cycles (20 min in total). Each sample was run in triplicate in each experiment. The change in fluorescence (Δfluorescence) was calculated as the fluorescence value at 20 min minus the fluorescence value at the beginning of the assay (0 min).

## Statistical analysis

We used GraphPad Prism (San Diego, CA, USA) for data analysis and for the drawing of figures. There were inevitable differences in the initial fluorescence values of the wells. We therefore calculated the increase in fluorescence values from baseline to the amplification plateau (at 20 min) for comparison. For determinations of positivity threshold during assay development, we performed $t$ tests to determine whether the mean Δfluorescence value of the standard positive control, containing a known concentration of plasmid DNA, was significantly different from that of the negative control (NC), for which nuclease-free water was added in place of the DNA. We considered $P$ values below 0.05 to be statistically significant.

For the E7 and L1 assays, we set different thresholds for the interpretation of positive and negative results following calculation of the Δfluorescence value obtained with an input of 100 copies/µL DNA, due to differences in the background values obtained in different assays (Fig. S1). The results for clinical samples were interpreted as follows: in

the E7 test, the result for a sample was considered positive if its mean Δfluorescence value was at least two times that of the NC; in the L1 or dual-detection tests, the result was considered positive if the Δfluorescence value was at least 10 times that of the NC. If the Δfluorescence for the sample was below these thresholds, the result was considered to be negative. The LOD was determined by elucidating the lowest concentration in which the HPV DNA template could be detected at or above 95% over a total of 20 replicates in 20 min. The sensitivity and specificity of detection systems for HPV16 and HPV18 were determined by comparison to the qPCR assay. The number of participants testing positive for HPV in the RPA-exo assay was divided by the number of participants testing positive for HPV in the qPCR assay to calculate the sensitivity. The number of participants testing negative for HPV in the RPA-exo assay was divided by the number of participants testing negative for HPV in the qPCR assay to calculate the specificity.

## ACKNOWLEDGMENTS

This project was supported by the Shanghai Municipal Science and Technology Major Project (Grant no. 2019SHZDZX02) and Shanghai International Science and Technology Cooperation Program (Grant no. 22490750200).

J.Y. and N.B. conceived and designed the experiments. J.Y. and L.M. designed primers and probes for RPA-exo and optimized the assays. J.Y., Y.T., and M.F. performed DNA extraction. W.F., Y.T., W.S., X.X., X.T., J.L., H.L., F.J., and Q.D. collected TCT and biopsy samples. J.Y. performed experiments. J.Y. and N.B. analyzed the data. J.Y., N.B., and G.W. prepared the manuscript. All the authors read and approved the final manuscript.

The authors declare no conflicts of interest.

## AUTHOR AFFILIATIONS

[1]Centre for Microbes, Development and Health, Institut Pasteur of Shanghai, Chinese Academy of Sciences, Unit of Discovery and Molecular Characterization of Pathogens, Shanghai, China

[2]Viral Hemorrhagic Fevers Research Unit, CAS Key Laboratory of Molecular Virology and Immunology, Institut Pasteur of Shanghai, Chinese Academy of Sciences, Shanghai, China

[3]University of the Chinese Academy of Sciences, Beijing, China

[4]Department of Obstetrics and Gynecology, Ruijin Hospital, Shanghai Jiao Tong University School of Medicine, Shanghai, China

[5]Institut Pasteur, Université Paris-Cite, Unité Environnement et Risque Infectieux, Cellule d'Intervention Biologique d'Urgence, Paris, France

[6]Institut Pasteur, Université Paris-cite, Unité Epidémiologie et Physiopathologie des Virus Oncogènes, Paris, France

## AUTHOR ORCIDs

Jiaxu Ying  http://orcid.org/0009-0000-8767-3379
Gary Wong  http://orcid.org/0000-0002-9044-8153
Weiwei Feng  http://orcid.org/0000-0002-0685-9479

## FUNDING

| Funder | Grant(s) | Author(s) |
| --- | --- | --- |
| Shanghai Municipal Science and Technology Major Project | 2019SHZDZX02 | Nicolas Berthet |
| Shanghai International Science and Technology Cooperation Program | 22490750200 | Gary Wong |

## ADDITIONAL FILES

The following material is available online.

### Supplemental Material

**Supplemental file 1 (Spectrum01207-23-s0001.docx).** Fig. S1 and Tables S1 to S4.

### Open Peer Review

**PEER REVIEW HISTORY (review-history.pdf).** An accounting of the reviewer comments and feedback.

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
