## [Reviewer comments · Microbiology Spectrum]

Microbiology Spectrum

Development and Validation of Real-time Recombinase Polymerase Amplification-Based Assays for Detecting HPV16 and HPV18 DNA

Jiaxu Ying, Lingjing Mao, Yujing Tang, Meriem Fassatoui, Wei Song, Xiaosheng Xu, Xiaojian Tang, Jing Li, Hua Liu, Fangfang Jian, Qinwen Du, Gary Wong, Weiwei Feng, and Nicolas Berthet

Corresponding Author(s): Nicolas Berthet, Institut Pasteur

Review Timeline:

Submission Date:	March 30, 2023
Editorial Decision:	May 11, 2023
Revision Received:	July 7, 2023
Editorial Decision:	August 4, 2023
Revision Received:	August 7, 2023
Accepted:	August 8, 2023

Editor: Peter Pelka

Reviewer(s): The reviewers have opted to remain anonymous.

Transaction Report:

DOI: <https://doi.org/10.1128/spectrum.01207-23>

May 11, 2023

Dr. Nicolas Berthet
Institut Pasteur
Virology
28 rue du docteur Roux
paris
France

Re: Spectrum01207-23 (Development and Validation of Real-time Recombinase Polymerase Amplification-Based Assays for Detecting HPV16 and HPV18 DNA)

Dear Dr. Nicolas Berthet:

Thank you for submitting your manuscript to Microbiology Spectrum. The reviewers found the manuscript of interest but have also highlighted certain improvements that can be made, which will strengthen the paper and I encourage you to incorporate them into a revised version.

Link Not Available

Sincerely,

Peter Pelka

Journals Department
Reviewer comments:

Reviewer #1 (Comments for the Author):

It was with interest that I read the original research article titled: "Development and Validation of Real-time Recombinase Polymerase Amplification-Based Assays for Detecting HPV16 and HPV18 DNA" by Ying et al. submitted to Microbiology Spectrum.

There is a great deal of activity towards developing sensitive and rapid molecular tests for high-risk HPV types, particularly using

isothermal techniques such as RPA and LAMP. There are notable advantages to these methods as they are rapid and cost-effective compared to many current approaches. Here, the authors describe an RPA assay paired with fluorogenic probes for the detection of HPV16 and HPV18 DNA. Both are high-risk types and leading causes of cervical cancers.

My major concern with this work is that there are multiple RPA assays for HPV that have already been published, many of which have been already paired with lateral flow or other more accessible visualization techniques. The authors mention that other RPA studies exist and provide citations but do not provide a strong justification as to why their assay is needed in addition to those already developed, they do not compare their results to any of these previous works, and they do not perform any side-by-side comparisons to existing RPA assays.

In addition, it is unclear where their 'gold standard' qPCR reference test is derived from and if it has been previously validated. Because of these factors, it feels like this study was performed 'in a vacuum' without consideration for outside works. As a result of this it becomes hard to compare their proposed solution for HPV testing to others being developed in the field.

Major comments:

The article describes the limit of detection (LOD) throughout; however, no statistics have been performed and the number of replicates is insufficient to measure LOD. A probit analysis would need to be performed. The limit of detection is defined as the amount of input material that would be required for a detection confidence of 95%. The authors should redesign their experiments to appropriately measure LOD or rephrase.

Here, comparisons were made to a 'gold standard' qPCR assay. Did this group design the primers for qPCR or are they commonly used for qPCR in the field. Please clarify. This is especially important as several samples that were identified as positive by clinical testing were not detected as positive in this study. This may indicate an issue with extraction efficiency or qPCR assay.

The authors make note that other RPA assays have been designed and published for HPV but they fail to discuss how their results compare to these previous works. Was a similar level of sensitivity and limit of detection observed? Why did they feel a new experimental design/primers were needed?

In the discussion, the authors should comment on their observed sensitivity and if this has implications on the diagnosis/detection of medically relevant cases.

The authors commonly refer to their assay as point-of-care, but there is still a great deal of work before this test could be used in a variety of settings - for example, how would the extraction steps be performed in resource-limited settings or by individuals with limited laboratory experience? This should be covered in the discussion.

Line 118: Wouldn't this tolerance to mismatches negatively impact specificity as well? The authors should mention this as a possibility, and includes alignments to other HPV types in Figure 1. The authors should also comment if there is a need for further specificity work against other HPV types.

Line 173: All tests were performed in triplicate; however, it is unclear how ambiguous results were interpreted as only one result is reported. If an RPA sample yielded two negatives and a positive would this be considered as a positive or negative result? Would this test be performed in triplicate in clinical settings as well? This information should factor into sensitivity calculations.

Minor comments:

Line 30 (Abstract): Cervical cancer is not the fourth-leading cause of death in women, please correct this statistic.

Line 41: For this line, please include a % sensitivity compared to qPCR instead of the LOD. The LOD is not a measure of sensitivity.

Line 63: Is not the leading cause of cancer deaths in 36 countries. It is the leading cause of female cancer deaths in 36 countries.

Line 102: Many different qPCR assays are very rapid due to breakthroughs in enzyme technologies and can be completed in <30 minutes. For example, the GeneXpert HPV test can perform both extraction and qPCR in <55 minutes.

Line 111: 'reduce' should read as 'reducing'.

Line 153/154: Why is an E7 plasmid being used to investigate specificity for the L1 target? Additionally, why are other HPV subtypes not being tested for specificity?

Line 168: Please define TCT at first use.

Line 211: Not all referenced assays require Cas12, at least one uses lateral flow for visualization of the HPV RPA reaction.

Line 212: You say 'most' do not detect multiple HPV types. Do some of these assays detect multiple types? Which ones?

Line 213-215: Consider moving the following statement to the introduction where it has more impact and provides context for the work: "The genotyping of HPV16 and HPV18 is recommended by the WHO as they are the genotypes most frequently implicated in patients with cervical cancer".

Line 252: For the samples not detected in this study but reported as positive at the hospital - is there an associated Ct value that was provided by the clinical test?

Line 264: Please list HPV types in numerical order.

Line 426: Please correct formatting on final line.

Figure 1: Please include alignments to several other HPV types and comment on whether off-target detection of other HPV types would be theoretically possible based on alignments.

Table 1: Can likely be moved to supplement.

Table 3: Please reformat table / fix spacing to make text easier to read and interpret.

Reviewer #2 (Comments for the Author):

The authors develop a cost-effective PCR-based assay to detect the two most common HPV types. Overall, I thought the paper was very well written and I have only a few minor comments:

- Page 7 line 143 - You only mention the LOD for the L1 plasmids, but have omitted the LOD for the E7 plasmids.
- If I understand correctly, the specificity of this assay was only tested with plasmids for type 16 and 18. This is a major limitation that should be mentioned in the discussion.
- page 10 line 217 - there is a typo "late flow strip (LFS)".
- page 11 line 239-242 " This finding indicates that the copy numbers of different HPV genes may be different for the same sample, with the clinical detection of only one gene segment potentially resulting in false-negatives results. " I do not understand this statement, can you please explain?
- Page 13, line 265 - "another genotype of HPV (66, 58)" The numbers in the brackets may be confused for references. I would suggest adding "type" or "genotype" to the numbers in the brackets.
- Page 13 line 265 "Sample 94 emitted both ROX and FAM fluorescence during dual detection (and was, thus, positive for both HPV16 and 18),"is it possible that there might be a co-infection with another type other than 16, given the specificity of the assay was only tested against type 16 and 18.
- Page 13 line 276- "only female cervical samples." Can you explain this?
- Page 13 line 283 - "which may infeasible or unacceptable" there is a typo.

Staff Comments:

Preparing Revision Guidelines

For complete guidelines on revision requirements, please see the journal Submission and Review Process requirements at

<https://journals.asm.org/journal/Spectrum/submission-review-process>. **Submissions of a paper that does not conform to Microbiology Spectrum guidelines will delay acceptance of your manuscript. "**

Please return the manuscript within 60 days; if you cannot complete the modification within this time period, please contact me. If you do not wish to modify the manuscript and prefer to submit it to another journal, please notify me of your decision immediately so that the manuscript may be formally withdrawn from consideration by Microbiology Spectrum.

Dear Dr. Pelka,

Thank you very much for giving us the opportunity to submit a revised draft of the manuscript “Development and Validation of Real-time Recombinase Polymerase Amplification-Based Assays for Detecting HPV16 and HPV18 DNA” for consideration of publication in *Microbiology Spectrum*. We appreciate the time and effort that you and the reviewers have dedicated to providing valuable feedback on our manuscript and are grateful for the insightful comments contributing to significant improvements to this study.

We have considered and incorporated the suggestions made by the reviewers. We have added replicates for each dilution to accurate the limit of detection, supplemented a comparison of various Recombinase Polymerase Amplification (RPA)-based methods (Table 1), as well as the references for qPCR primers and probes. We have also used different DNA extraction methods to validate the assays (Table S4). Improvements were also made to address some layout issues. All changes are highlighted within the tracked-changes version of the manuscript. Below are our point-by-point responses to reviewer comments and concerns.

Thank you again for all your comments and suggestions. We look forward to hearing from you. Please do not hesitate to contact us with any further questions and comments that you may have.

Yours sincerely,

Nicolas Berthet, Pharm.D, Ph.D

Institut Pasteur, Université Paris-cite,

Unité Epidémiologie et Physiopathologie des Virus Oncogènes

Review #1

Major comments:

The article describes the limit of detection (LOD) throughout; however, no statistics have been performed and the number of replicates is insufficient to measure LOD. A probit analysis would need to be performed. The limit of detection is defined as the amount of input material that would be required for a detection confidence of 95%. The authors should redesign their experiments to appropriately measure LOD or rephrase.

Response: Thank you for your suggestion. We have increased the number of replicates for each diluted concentration of the plasmid to 20 to more accurately determine the LOD. The LOD of assays were determined by elucidating the lowest concentration in which the HPV plasmid template could be detected at or above 95% over a total of 20 replicates (Figure 1b).

This sentence was added in the Statistical analysis section: “The LOD was determined by elucidating the lowest concentration in which the HPV DNA template could be detected at or above 95% over a total of 20 replicates in 20 min.”

Here, comparisons were made to a 'gold standard' qPCR assay. Did this group design the primers for qPCR or are they commonly used for qPCR in the field. Please clarify. This is especially important as several samples that were identified as positive by clinical testing were not detected as positive in this study. This may indicate an issue with extraction efficiency or qPCR assay.

Response: Thank you for pointing out this issue. We did not correctly cite the source of the qPCR primers used. In the revised version, we have added references for the qPCR primers and probes in Table 2. In this article, we also used serially diluted plasmids to construct standard curves for HPV16/18 using these two sets of primers and probes, with correlation coefficients greater than 0.99, which also confirms the feasibility of these qPCR systems. Partial inconsistency between our qPCR results and clinical testing may be due to slight differences in the sampling points or low DNA concentrations.

The authors make note that other RPA assays have been designed and published for HPV but they fail to discuss how their results compare to these previous works. Was a similar level of sensitivity and limit of detection observed? Why did they feel a new experimental design/primers were needed?

Response: Thank you very much for your suggestion. We have added a comparison of currently developed RPA based HPV detection methods with our system in various aspects to the article (Table 1). Our method is the first real-time RPA detection method for HPV16/18 using exo probes, reaching a similar level of sensitivity compared with other methods. Some existing methods are based on microfluidics, which are still difficult to implement in the clinic, while fluorescence collectors are still widely used in field and clinical laboratories.

This sentence was added in the Introduction section: “Until now, a total of 15 articles have applied RPA to HPV detection (Table 1).”

This sentence was added in the Discussion section: “Considering the detection section alone, our assay has advantages of easy primer/probe design, quick operation, short incubation time, constant and low incubation temperature, and genotyping in 1 reaction.”

In the discussion, the authors should comment on their observed sensitivity and if this has implications on the diagnosis/detection of medically relevant cases.

Response: Thank you for pointing this out. We have revised the discussion of the sensitivity in lieu of Ct values from qPCR, the revised text reads as follows: “As stated above, the HPV16 L1 RPA-exo assay, detected viral DNA with Ct values below 34.3, equivalent to 26.5 copies/ μ L, according to the standard curve. The HPV18 L1 single RPA-exo assay detected viral DNA with a values below 34.4, equivalent to 12.6 copies/ μ L. The dual-detection system was also able to detect Ct values as low as 34.3 for HPV16 (patient 106’s TCT) and 34.4 for HPV18 (patient 68’s TCT). The presence of more than 10 copies/ μ L of HPV16 or HPV18 genome is considered clinically significant, and lower values probably reflect transient infections.”

The authors commonly refer to their assay as point-of-care, but there is still a great deal of work before this test could be used in a variety of settings - for example, how would the extraction steps be performed in resource-limited settings or by individuals with limited laboratory experience? This should be covered in the discussion.

Response: Thank you for pointing out this problem. We agree that this is a limitation of this study. These clinical samples are precious and limited in quantity, and were also used for other applications not related to this study. We needed to get long fragments of DNA as much as possible, so we chose the phenol-chloroform method instead of commercial kits for DNA extraction.

We have added this as a limitation and the revised text reads as follows: “This study had several limitations, including its small sample size, the use of only female cervical samples (no plasma samples), did not use other HPV genotypes for specificity verification and tedious phenol-chloroform sample extraction method.

However, this detection system is applicable to all DNA samples, which means commercial kits can be used for DNA extraction as well.”

To validate that DNA extracted by other methods are also applicable for our RPA-exo assays, we used the Thermo GeneJet Viral DNA and RNA Purification Kit to extract DNA from 20 TCT samples and crude DNA extraction for 3 TCT samples. The results are presented in Table S4, and the methods have been added in the corresponding section as follows: “DNA of 20 TCT samples were also extracted by Thermo GeneJet Viral DNA and RNA Purification Kit (Thermo Fisher Scientific, Waltham, MA, USA) following the manufacturer’s instructions. Crude DNA preparation was used for 3 TCT samples. The cells were suspended in 100 μ L lysis solution (Thermo Fisher Scientific, Waltham, MA, USA) with proteinase K and incubated at 56°C for 15 min, then the proteinase K was heat inactivated.”

The comment is also added in the Results section and reads as follows: “There was also good agreement in clinical samples using other DNA extraction methods, validating the applicability of these assays for DNA with multiple extraction methods (Table S4).”

Line 118: Wouldn't this tolerance to mismatches negatively impact specificity as well? The authors should mention this as a possibility, and includes alignments to other HPV types in Figure 1. The authors should also comment if there is a need for further specificity work against other HPV types.

Response: Thank you for your comment. The reviewer is correct that the tolerance to mismatches negatively impact the specificity. The sensitivity of single detection (1,000 copies) is better than that of dual detection (10,000 copies) when tested with plasmids. We have redrawn Figure 2 to show the alignment of 15 high-risk HPVs. As shown in the figure, our degenerate primers target conserved regions, so the

possibility of amplifying other genotypes cannot be ruled out. However, we used specific HPV16/18 probes, in which the target regions are highly divergent. Therefore, we did not use other HPV genotypes to verify the specificity.

The comment is added in the Discussion section and reads as follows: “Due to the lack of DNA templates for other HR-HPVs, we only used HPV18 DNA when verifying the specificity of HPV16 detection systems, and vice versa. The single genotype assays were all considered to be highly specific, as specific primers and probes were used. For the dual-detection system, the alignments showed that our probes targeted the highly divergent region (Figure 2). As shown, the degenerate primers had the potential to amplify other genotypes, but the specific probes eliminated the possibility of off-target detection, as the region to which the probes bind were not conserved. The designed probe had <90% nucleotide sequence similarity with any other HR-HPV types in the interval, thus ensuring the accurate identification of the corresponding HPV type. When we used this system to detect samples that were only HPV16 positive, there was no fluorescence detected from the HPV18 probe (and vice versa), demonstrating the specificity of the dual-detection assay.”

Line 173: All tests were performed in triplicate; however, it is unclear how ambiguous results were interpreted as only one result is reported. If an RPA sample yielded two negatives and a positive would this be considered as a positive or negative result? Would this test be performed in triplicate in clinical settings as well? This information should factor into sensitivity calculations.

Response: Thank you very much for your comments. In the clinical setting, this test will also be performed in triplicate. We also used defined thresholds ($L1 > 10$ times NC and $E7 > 2$ times NC) to determine whether the sample is positive or not. If the sample produced two negative results, and one positive result, it would be considered

negative, whereas two positive and one negative result would be considered positive. For some samples with Δ fluorescence value near the detection borderline, we did more replicates to increase confidence in the results.

We have added this sentence in the table legend of Table S2 and Table S3: “+ or – indicate consistent results between all replicates. The numbers in parentheses indicate the number of positive results out of the total number of replicates.”

Minor comments:

Line 30 (Abstract): Cervical cancer is not the fourth-leading cause of death in women, please correct this statistic.

Response: Thank you for pointing this out. The source of this statistic is from the GLOBOCAN 2020 database, and we revised it to be more precise.

The revised text reads as follows: “Cervical cancer is the fourth-leading cause of cancer-related deaths among women worldwide.”

Line 41: For this line, please include a % sensitivity compared to qPCR instead of the LOD. The LOD is not a measure of sensitivity.

Response: Thank you for your suggestion, we used the comparison of Ct value from qPCR to indicate the sensitivity. The line reads as follows: “These assays were highly sensitive, able to detect clinical samples with a Ct value below 34, and yielded results within 25 minutes.”

Line 63: Is not the leading cause of cancer deaths in 36 countries. It is the leading cause of female cancer deaths in 36 countries.

Response: Thank you for your correction. The line now reads as follows: “the leading cause of female cancer deaths in 36 countries”.

Line 102: Many different qPCR assays are very rapid due to breakthroughs in enzyme technologies and can be completed in <30 minutes. For example, the GeneXpert HPV test can perform both extraction and qPCR in <55 minutes.

Response: Thank you for pointing out this issue. GeneXpert is a truly convenient point-of-care device, which has popularized HPV testing in low-income and middle-income countries. However, in clinical laboratories with limited resources and a large number of samples to process, using such an instrument is inefficient and costly. The aim of these isothermal rapid detection systems is not to replace qPCR methods but to reduce the burden of qPCR in local clinical laboratories, simplify experimental operations, and reduce costs.

We have made the following corrections and the revised text reads as follows: “However, these techniques require expensive specialist equipment to control the temperature cycle, and different companies always require corresponding equipment for conducting experiments with their kits and results readout, resulting in an overall increase in the cost of the tests.”

Line 111: 'reduce' should read as 'reducing'.

Response: We have made the requested change.

Line 153/154: Why is an E7 plasmid being used to investigate specificity for the L1 target? Additionally, why are other HPV subtypes not being tested for specificity?

Response: Thank you for pointing this out. The HPV18 E7 plasmid was used to investigate the specificity for HPV16 E7 RPA-exo detection. The correction has been made and the line now reads as follows: “For example, for the HPV16 E7 RPA-exo detection, we used 10^4 copies/ μL of the HPV18 E7 plasmid to check the specificity”.

The reason for why we did not test for specificity against other HPV subtypes has

been discussed in the text: “Due to the lack of DNA templates for other HR-HPVs, we only used HPV18 DNA when verifying the specificity of HPV16 detection systems, and vice versa. The single genotype assays were all considered to be highly specific, as specific primers and probes were used. For the dual-detection system, the alignments showed that our probes targeted the highly divergent region (Figure 2). As shown, the degenerate primers had the potential to amplify other genotypes, but the specific probes eliminated the possibility of off-target detection, as the region to which the probes bind were not conserved. The designed probe had <90% nucleotide sequence similarity with any other HR-HPV types in the interval, thus ensuring the accurate identification of the corresponding HPV type. When we used this system to detect samples that were only HPV16 positive, there was no fluorescence detected from the HPV18 probe (and vice versa), demonstrating the specificity of the dual-detection assay.”

Line 168: Please define TCT at first use.

Response: Done. TCT stands for “ThinPrep cytological test”.

Line 211: Not all referenced assays require Cas12, at least one uses lateral flow for visualization of the HPV RPA reaction.

Response: Thank you for pointing this out. Based on Table 1, we have made modifications to this sentence to be more accurate. The revised text reads as follows: “For RPA based HPV detection assays, over half of the systems are still combined with Cas12. Out of a total of 15 assays, 9 used Cas12 (Table 1)”

Line 212: You say 'most' do not detect multiple HPV types. Do some of these assays detect multiple types? Which ones?

Response: Thank you for pointing this out. Based on the Table we have compiled (Table 1), we have made modifications to this sentence. The revised text reads as follows: “In addition, most (8 out of 15) of these assays (Table 1) can only achieve

separate genotyping, in which the detection of different genotypes still needs to be achieved in different reaction systems, and multiple sets of primers or probes cannot be combined into one reaction. For the assay developed in this study, genotyping in the same reaction has the advantages of saving reagent costs and reducing the duplicate addition of samples.”

Line 213-215: Consider moving the following statement to the introduction where it has more impact and provides context for the work: "The genotyping of HPV16 and HPV18 is recommend by the WHO as they are the genotypes most frequently implicated in patients with cervical cancer".

Response: Done.

Line 252: For the samples not detected in this study but reported as positive at the hospital - is there an associated Ct value that was provided by the clinical test?

Response: Unfortunately, the Ct values from all the samples were not available from our collaborators at the hospital. We also added this sentence in the Discussion. “Unfortunately, the Ct value of all the samples was not available from our collaborators at the hospital.”

Line 264: Please list HPV types in numerical order.

Response: Done.

Line 426: Please correct formatting on final line.

Response: Done.

Figure 1: Please include alignments to several other HPV types and comment on whether off- target detection of other HPV types would be theoretically possible based on alignments.

Response: Thank you for your suggestion, we have now added alignments to other HR-HPVs in Figure 2 (formerly Figure 1). The Figure legend was revised and reads

as follows: “The GenBank accession numbers for the sequences used are as follows: K02718.1 (HPV16), X05015.1 (HPV18), J04353.1 (HPV31), M12732.1 (HPV33), X74477.1 (HPV35), LR861938.1 (HPV39), X74479.1 (HPV45), M62877.1 (HPV51), X74481.1 (HPV52), X74483.1 (HPV56), D90400.1 (HPV58), X77858.1 (HPV59), DQ080079.1 (HPV68), X94165.1 (HPV73), and AB027021 (HPV82). The alignment was performed by the Geneious Prime program and shown in SnapGene. The consensus sequence with threshold of >90% is shown in yellow highlight. The degenerate primers and specific probes (modifications not marked) for dual RPA-exo detection are indicated by different colored boxes. The forward primer is shown above the red box, and the reverse primer is shown above the blue box. When choosing degenerate bases, only HPV16 and HPV18 genotypes were considered, marked with small boxes of corresponding colors. The HPV16 L1 RPA exo probe is shown in the green box while the HPV18 L1 RPA exo probe is shown in the orange box.”

This sentence was also added in the Discussion section: “As shown, the degenerate primers have the potential to amplify other genotypes, but the specific probes eliminate the possibility of off-target detection, as the region to which the probes bind is not conserved.”

Table 1: Can likely be moved to supplement.

Response: Thank you for your suggestion, we have moved this Table to the supplemental material as Table S1.

Table 3: Please reformat table / fix spacing to make text easier to read and interpret.

Response: Thank you for your suggestion. We have reformatted all of the Tables to make them easier to read.

Reviewer #2:

Minor comments:

•Page 7 line 143 - You only mention the LOD for the L1 plasmids, but have omitted the LOD for the E7 plasmids.

Response: Thank you for pointing this out. We did not design detection systems for HPV16/18 E7 qPCR, so all assays are compared with L1 qPCR systems. For single HPV16 E7 RPA-exo and single HPV18 E7 RPA-exo, the limit of detection is 100 copies/ μ L, in which the threshold was defined as the Δ fluorescence of DNA being equal to or higher than twice than that of the NC. Most of the existing HPV DNA detection methods targets L1, as it is the basis for HPV genotyping, while E7 or E6 DNA detection are less applied in HPV screening. One of the problems for designing E6/E7 detection systems is the high %GC in this region, which may result high melting temperature of the primers and thus hard to anneal the template. This phenomenon was also found in our assays, may lead to the high background fluoresce and reported in Reference 51 of the revised manuscript. Therefore, we did not design E7 qPCR systems and choose L1 qPCR as the standard instead.

We explained the reason for choosing E7 in the Discussion section and described the differences found using these two systems: “When the E7 protein sequence was used to construct a phylogenetic tree, it was found that oncogenic or possibly oncogenic HPVs clustered together. As the E7 expressed by HR-HPVs can degrade pRb, an important tumor suppressor factor, we thought that the use of E7 as the target gene may be more clinically meaningful. As the target gene for E7 RPA-exo assays was different from the qPCR target, the results were less consistent. For example, the samples from patient 153 (T153 and B153) and the TCT sample from patient 167 (T167) gave positive results for L1 detection, but the E7 tests were negative, whereas the opposite was observed for the biopsy specimen from patient 79 (B79).”

- If I understand correctly, the specificity of this assay was only tested with plasmids for type 16 and 18. This is a major limitation that should be mentioned in the discussion.

Response: Thank you for pointing out this issue. We have added a comment about the specificity test and the text now reads as follows: “Due to the lack of DNA templates for other HR-HPVs, we only used HPV18 DNA when verifying the specificity of HPV16 detection systems, and vice versa. The single genotype assays were all considered to be highly specific, as specific primers and probes were used. For the dual-detection system, the alignments showed that our probes targeted the highly divergent region (Figure 2). As shown, the degenerate primers had the potential to amplify other genotypes, but the specific probes eliminated the possibility of off-target detection, as the region to which the probes bind were not conserved. The designed probe had <90% nucleotide sequence similarity with any other HR-HPV types in the interval, thus ensuring the accurate identification of the corresponding HPV type. When we used this system to detect samples that were only HPV16 positive, there was no fluorescence detected from the HPV18 probe (and vice versa), demonstrating the specificity of the dual-detection assay.”

- page 10 line 217 - there is a typo "late flow strip (LFS)".

Response: Thank you for pointing this out. This has now been corrected to “lateral flow strips”.

- page 11 line 239-242 " This finding indicates that the copy numbers of different HPV genes may be different for the same sample, with the clinical detection of only one gene segment potentially resulting in false-negatives results. " I do not understand this statement, can you please explain?

Response: The meaning of this sentence is that HPV can integrate into the host genome, leading to the interruption and inactivation of viral genes. If the PCR primers

used happen to be located around the break sites, it may fail to amplify, resulting in false negative results. The observation in our experiment that L1 successfully amplifies but E7 fails in the same sample (or E7 successfully amplifies but L1 cannot be detected) can be explained by the description above.

We have modified the statement and the revised text reads as follows: “This finding indicates that for the same sample, the detection results for different gene regions of HPV may differ, and the clinical detection of only one gene segment may potentially result in false-negative outcomes.”

- Page 13, line 265 - "another genotype of HPV (66, 58)" The numbers in the brackets may be confused for references. I would suggest adding "type" or "genotype" to the numbers in the brackets.

Response: Thank you for your suggestion, we have revised the text as “another genotype of HPV (genotype 58 and 66)”.

- Page 13 line 265 "Sample 94 emitted both ROX and FAM fluorescence during dual detection (and was, thus, positive for both HPV16 and 18),"is it possible that there might be a co- infection with another type other than 16, given the specificity of the assay was only tested against type 16 and 18.

Response: Thank you for your comments. The kits used in the clinical laboratory can detect 15 HR-HPV types (HPV16, 18, 31, 33, 35, 39, 45, 51, 52, 56, 58, 59, 66, 68, 82). Since there were no other HR-HPV infections in the information provided by the hospital, we believe that the results are reliable. For Sample 94, the hospital only marked it as HPV18 positive, but it was shown to be co-infected with HPV16 with our qPCR and RPA-exo systems. In the manuscript, we pointed out that some samples labeled as HPV16 or HPV18 were co-infected with other HPVs, but only fluorescence

corresponding to HPV16/18 was detected, which indirectly supports the specificity of RPA-exo dual-detection.

- Page 13 line 276- "only female cervical samples." Can you explain this?

Response: Thank you your comments. In addition to routine cervical swab samples, we also sought to verify whether our detection systems can be used to detect circulating HPV DNA (HPV ctDNA) in plasma, which can be a meaningful study, as HPV ctDNA may serve as a biomarker for cervical cancer surveillance.

We revised the text as follows: “the use of only female cervical samples (i.e. no plasma samples)”

- Page 13 line 283 - "which may infeasible or unacceptable" there is a typo.

Response: Thank you for pointing this out. The correction has been made and the text reads as follows: “which may be unfeasible or unacceptable”.

August 4, 2023

Dr. Nicolas Berthet
Institut Pasteur
Virology
28 rue du docteur Roux
paris
France

Re: Spectrum01207-23R1 (Development and Validation of Real-time Recombinase Polymerase Amplification-Based Assays for Detecting HPV16 and HPV18 DNA)

Dear Dr. Nicolas Berthet:

Thank you for submitting your manuscript to Microbiology Spectrum. As you will see your paper is very close to acceptance. Please modify the manuscript along the lines the reviewer has recommended. As these revisions are quite minor, I expect that you should be able to turn in the revised paper in less than 30 days, if not sooner. If your manuscript was reviewed, you will find the reviewers' comments below.

When submitting the revised version of your paper, please provide (1) point-by-point responses to the issues raised by the reviewers as file type "Response to Reviewers," not in your cover letter, and (2) a PDF file that indicates the changes from the original submission (by highlighting or underlining the changes) as file type "Marked Up Manuscript - For Review Only". Please use this link to submit your revised manuscript. Detailed instructions on submitting your revised paper are below.

Link Not Available

Sincerely,

Peter Pelka

Reviewer comments:

Reviewer #1 (Comments for the Author):

I thank the authors for taking time to address all of my major comments, and I am satisfied with the corrections. In addition, I have the following minor comments.

Minor Comments:

Line 41: Should likely read, 'able to detect clinical samples with Ct values above 34...' or "was able to detect HPV in all clinical samples with CT values below 34".

Line 47 should read: "... advantages of yields results rapidly and operating at a constant temperature, while being cost effective and easy to use."

Line 65: China has the highest number of cervical cancer deaths, but also is essentially tied for the largest population. Because of this, it doesn't make sense to compare the rates of cervical cancer in China to the rest of the world by sheer number alone- the statistic has little weight and doesn't reflect the actual burden compared to other countries unless it is corrected to per capita. Just focus on China alone and state the number of total cases and the impact on China, or correct to per capita to compare rates to other countries globally.

Line 66: Again, this statistic doesn't have much weight as China and India account for 1/3 of the world's population - it stands to reason that they would have approximately 1/3 of the world's cancer burden.

Line 70-72: Please avoid long URLs in text and modify to be a citation.

Line 76: to avoid confusion with a citation, please include the word Types in the parentheses when referring to HPV types. For example, multiple types of HPV (Types 3, 4, and 7) were found to be...

Preparing Revision Guidelines

Please return the manuscript within 60 days; if you cannot complete the modification within this time period, please contact me. If you do not wish to modify the manuscript and prefer to submit it to another journal, please notify me of your decision immediately so that the manuscript may be formally withdrawn from consideration by Microbiology Spectrum.

Dear Dr. Pelka,

Thank you very much for giving us the opportunity to submit a revised draft of the manuscript “Development and Validation of Real-time Recombinase Polymerase Amplification-Based Assays for Detecting HPV16 and HPV18 DNA” for consideration of publication in *Microbiology Spectrum*. We are grateful to the time and effort that you and the reviewers have dedicated to reviewing our manuscript and providing valuable comments. We have carefully considered and responded to each suggestion.

All changes are highlighted within the tracked-changes version of the manuscript. Below are our point-by-point responses to reviewer comments.

Thank you again for all your suggestions and advice. We hope that these modifications can fulfill the requirements for manuscript acceptance. Please do not hesitate to contact us with any further questions and comments that you may have.

Yours sincerely,

Nicolas Berthet, Pharm.D, Ph.D

Institut Pasteur, Université Paris-cite,

Unité Epidémiologie et Physiopathologie des Virus Oncogènes

E-mail: nicolas.berthet@pasteur.fr

Reviewer #1

Minor comments:

Line 41: Should likely read, 'able to detect clinical samples with Ct values above 34...' or "was able to detect HPV in all clinical samples with CT values below 34".

Response: Done.

Line 47 should read: "... advantages of yields results rapidly and operating at a constant temperature, while being cost effective and easy to use."

Response: Done.

Line 65: China has the highest number of cervical cancer deaths, but also is essentially tied for the largest population. Because of this, it doesn't make sense to compare the rates of cervical cancer in China to the rest of the world by sheer number alone- the statistic has little weight and doesn't reflect the actual burden compared to other countries unless it is corrected to per capita. Just focus on China alone and state the number of total cases and the impact on China, or correct to per capita to compare rates to other countries globally.

Response: Thank you very much for your suggestion, we do need to consider China's large population base before comparison. After modification, the revised text is as follows:

“In 2020, China reported 109,741 cases of cervical cancer and 59,060 deaths, with a corresponding age-standardized incidence of 10.7 cases per 100,000 women-years and mortality rate of 5.3 deaths per 100,000 women-years. The incidence of cervical cancer in China is currently still above the threshold set by the World Health Organization (WHO)’s Cervical Cancer Elimination Initiative, which is 4 cases per 100,000 women-years.”

Institut Pasteur

Line 66: Again, this statistic doesn't have much weight as China and India account for 1/3 of the world's population - it stands to reason that they would have approximately 1/3 of the world's cancer burden.

Response: Thank you for your suggestion, we deleted this sentence and focused on the data in China. The revised text has shown in the previous response.

Line 70-72: Please avoid long URLs in text and modify to be a citation.

Response: Done

Line 76: to avoid confusion with a citation, please include the word Types in the parentheses when referring to HPV types. For example, multiple types of HPV (Types 3, 4, and 7) were found to be...

Response: Thank you for your suggestion. We added "genotype" in the parentheses to make the context more clear and consistent.

August 8, 2023

Dr. Nicolas Berthet
Institut Pasteur
Virology
28 rue du docteur Roux
paris
France

Re: Spectrum01207-23R2 (Development and Validation of Real-time Recombinase Polymerase Amplification-Based Assays for Detecting HPV16 and HPV18 DNA)

Dear Dr. Nicolas Berthet:

Your manuscript has been accepted, and I am forwarding it to the ASM Journals Department for publication. You will be notified when your proofs are ready to be viewed.

Sincerely,

Peter Pelka
Editor, Microbiology Spectrum
